# Trends in MitraClip Placements and Predictors of 90-Day Heart Failure Rehospitalization: A Nationwide Analysis

**DOI:** 10.3390/medsci13030081

**Published:** 2025-06-20

**Authors:** Vivek Joseph Varughese, Vignesh Krishnan Nagesh, Seetharamaprasad Madala, Ruchi Bhuju, Carra Lyons, Simcha Weissman, Adam Atoot, Dominic Vacca, Budoor Alqinai

**Affiliations:** 1Department of Internal Medicine, University of South Carolina, Prisma Health, Columbia, SC 29201, USA; seetharamprasad.madala@prismahealth.org (S.M.); carra.powell@prismahealth.org (C.L.); dominic.vacca@prismahealth.org (D.V.); 2Department of Internal Medicine, Hackensack Palisades Medical Center, North Bergen, NJ 07047, USA; vgneshkrishnan@gmail.com (V.K.N.); ruchibhuju3@gmail.com (R.B.); simchaweissman@gmail.com (S.W.); adamatoot.md@gmail.com (A.A.);

**Keywords:** MitraClip, heart failure readmissions, functional mitral regurgitation

## Abstract

**Background:** Chronic mitral regurgitation (MR) is categorized into primary and secondary MR (SMR). While primary MR arises from structural abnormalities of the mitral valve apparatus, SMR is a consequence of cardiac remodeling, typically due to heart failure or atrial fibrillation. Management strategies differ significantly, with primary MR requiring direct valvular intervention and SMR necessitating a comprehensive approach incorporating guideline-directed medical therapy (GDMT), revascularization, and resynchronization strategies. The MitraClip, a transcatheter edge-to-edge repair (TEER) device, has emerged as a recommended intervention for symptomatic severe SMR despite optimal GDMT. **Objectives**: This study aims to evaluate national trends in MitraClip placements in the U.S. from 2016 to 2021 and to assess 90-day readmission events following the procedure. Additionally, we analyze patient and socioeconomic factors associated with heart failure readmissions post-MitraClip placement to optimize patient selection criteria. **Methods**: The study utilized data from the National Inpatient Sample (NIS) for the years 2016–2021 and the National Readmissions Database (NRD) for 2021. Patients who underwent MitraClip placement were identified using ICD-10 code 02UG3JZ. We stratified the population based on demographics, hospital resource utilization, and comorbidities. Index admissions were classified based on the presence or absence of heart failure remissions within 90 days post-procedure. Statistical analyses, including ANOVA and logistic regression, were conducted to identify factors associated with readmissions. **Results**: MitraClip utilization demonstrated a rising trend from 2016 to 2021, with total annual procedures increasing from 869 to 2488. Mean patient age remained stable at 76–79 years, with a nearly equal sex distribution. In-hospital mortality remained low (1–3%) throughout the study period. A steady increase in hospital charges was observed, alongside a decline in the mean length of stay. Analysis of 4918 index admissions for MitraClip placement in 2021 identified 780 total readmissions within 90 days, with 206 (26.4%) attributed to heart failure. Factors significantly associated with increased risk of heart failure readmissions included atrial fibrillation (OR 3.77, CI 1.82–4.23), pulmonary hypertension (OR 3.96, CI 1.49–5.55), and chronic lung disease (OR 1.91, CI 1.32–2.77). **Conclusions**: The increasing adoption of MitraClip underscores its growing role in managing SMR. However, heart failure readmissions remain a significant concern. Identifying high-risk patient profiles can refine selection criteria and enhance post-procedural management strategies to improve clinical outcomes. Further research is needed to optimize patient selection and refine risk stratification for MitraClip interventions.

## 1. Introduction

Chronic Mitral regurgitation can be broadly subdivided into Primary and Secondary Mitral Regurgitation (SMR). While primary mitral regurgitation occurs from defects associated with the mitral valve apparatus, i.e., the leaflets, annulus, papillary muscles, or chordae tendineae, chronic secondary mitral regurgitation is considered a primary pathology of the left atrium and the ventricle. Secondary mitral valve regurgitation stems from defects in coaptation of the mitral leaflets due to cardiac remodeling. This could either be from wall motion abnormalities caused due to prior myocardial infarction, cardiac remodeling because of long-standing heart failure, or left atrial dilatation caused by long standing atrial fibrillation. The difference they hold in etiology also reflects in the management strategies as well as the long-term outcomes. Primary MR is essentially a structural defect of the mitral valve apparatus, and management relies on the correction, often surgical, of the valve apparatus. However, in secondary mitral regurgitation, regurgitation is only one part of the pathology. Looking at the current AHA/ACC guidelines on the management of valvular heart disease, management of secondary MR is recommended to be under a team approach, with a heart failure specialist being an integral part of the team. Secondary mitral regurgitation is often reversible through reverse remodeling achieved through guideline-directed medical therapy, revascularization, and resynchronization strategies. This becomes even more evident looking at the guidelines for the management of each of the conditions [1]. In symptomatic severe primary MR, as well as asymptomatic severe MR with evidence of LV dysfunction (defined as Ejection fraction < 60% or LVEDD < 4 cm), recommendations are for mitral valve surgery or repair with Transcatheter Edge-to- Edge Repair (TEER) recommended only in non-surgical candidates. GDMT holds a role in patients who are not candidates for interventions or while waiting for the intervention. While in the management of secondary MR, interventions like valvular replacement, repair, or TEER are only recommended in severe symptomatic MR with persistent symptoms despite optimal GDMT and revascularization strategies. In fact, the current AHA guidelines define severe symptomatic secondary MR as persistent symptoms despite revascularization and remodeling strategies in addition to the echocardiographic parameters defining severity (EROA > 0.4 cm^2^, regurgitant volume > 60 mL, or regurgitant fraction > 50%). While repair holds noticeable superiority in outcomes compared to replacement in primary MR, such a distinction is not observed in secondary MR, further proving differences between these two entities, in etiology as well as management strategies. In other words, while primary MR is essentially a structural pathology associated with the mitral valve apparatus, secondary MR is more of a consequence of cardiac remodeling [2].

MitraClip is a transcatheter approach to mechanically fix the malcoaptation of mitral leaflets caused due to annular dilatation in secondary MR. It falls under the category of Transcatheter Edge-to-Edge repair of mitral valve apparatus, and is a recommended intervention strategy for severe symptomatic secondary MR. Current guidelines recommend it as a treatment modality in severe symptomatic secondary MR with persistent symptoms despite being on maximally tolerated GDMT, having an EF between 20% and 50%, given that the Pulmonary Artery Systolic Pressure (PASP) remains less than 70 mm Hg and Left Ventricular End Systolic Dimension (LVESD) remains less than 7 cm. Surgical mitral valve repair or replacement is recommended in patients who are not candidates for TEER (MitraClip). This puts MitraClip as more of a management strategy for advanced heart failure rather than a structural intervention for a primary valve disease. As mentioned before, secondary MR is a consequence of cardiac remodeling due to advanced heart failure, and at the same time is also a factor worsening cardiac remodeling, creating a vicious cycle. This makes the management challenging, often creating a gray area between the switch from medical management to interventions. This is analogous to the situation of ischemic cardiomyopathy management, where trials like the STITCH [3] compared revascularization strategies (CABG) to medical management, and COURAGE [4] compared PCI with GDMT to GDMT alone in stable ischemic heart disease. There have been trials like COAPT and the MITRA AF [5] to optimize patient selection for TEER. While the COAPT [6] trial demonstrated better outcomes in terms of heart failure hospitalization and death in 24 months in the TEER plus GDMT group compared to GDMT alone, the MITRA AF trial showed no difference in outcomes between the groups. However, the extent of cardiac remodeling in patients who were included in the MITRA AF trial was substantially higher, including patients with LVESD > 7 cm and a lower mean EROA (0.3 cm^2^) compared to patients included in the COAPT trial (0.4 cm^2^). The RESHAPE- HF2 trial [7] proved that there were superior outcomes in the TEER plus GDMT group in terms of one-year heart failure hospitalizations and death, as well as improvement from the baseline KCCQ scores. The EVEREST [8] trial proved surgical repair/replacement to be superior in terms of heart failure hospitalizations and death when compared to TEER, but with higher postoperative complications, hospital stay, and hospital charges. The definition of the “perfect patient” for the MitraClip still remains a gray area. With advanced heart failure strategies, including assist devices, continuous inotropic infusions, and heart transplantations, turning more robust and prevalent, the selection of patients having a meaningful difference in heart failure symptoms through TEER becomes relevant. The aim of our study is to review the national trends in MitraClip placements in the US between the years 2016 and 2021, as well as the 90-day readmission events following MitraClip (TEER) procedures, and analyze the readmissions for heart failure in the 90 days following discharge. Patient as well as socioeconomic factors that held significant association with heart failure readmissions after the procedure were studied to provide insights into optimal patient selection for the procedure.

## 2. Methods

The National Inpatient Sample (NIS) database for the years 2016–2021 was used for the population analysis. The National Readmissions Database (NRD) 2021 was used to analyze the 90-day readmission events following MitraClip procedures. NIS and NRD are the largest publicly available databases in the United States (US), and all the data involved were de-identified.

ICD (International Classification of Diseases) 10 code 02UG3JZ was used to select patients who underwent MitraClip (transcatheter edge-to-edge repair) in both NIS and NRD [9]. Population stratification was performed for age, sex, race, and household income. Mean length of hospital stay and mean of total hospital charges were also calculated over the years 2016–2021. The study population was stratified based on non-rheumatic mitral valve insufficiency (functional MR) using the ICD 10 code I340 and only MitraClip procedures with documented non-rheumatic mitral valve insufficiency were included in the study. Admissions with diagnosis of mitral valve prolapse (ICD 10 code I341) were excluded. In the NRD, we stratified index admissions for MitraClip procedures into two groups based on the recorded readmission events for heart failure in the 90 days following discharge. Heart Failure readmissions were selected using admitting diagnosis (I10_dx1): I110, I120, I130, I1320. One-way ANOVA was used to stratify the differences in the prevalence of comorbidities in both the groups (Kolmogorov–Smirnov test). A two-tailed *p* value < 0.05 was used to determine statistical significance of difference in comorbidities. Comorbidities that held significant differences between the groups were analyzed using logistic regression, analyzing the association with heart failure readmissions in the 90 days following discharge. Multivariate regression analysis (probit) was used for comorbidities that held significant association with heart failure readmissions, after accounting for age, sex, race, monthly income, and comorbidities that held significant differences in the ANOVA. *p* value < 0.05 was used to determine statistical significance. Age, Sex, median quarterly income, APDRG risk severity indices, Charleson comorbidity index, and factors that had significant association in the univariate analysis were included in the study.

## 3. Results

The trends as well as sociodemographic parameters for MitraClip procedures are summarized in Table 1.

Analyzing the total number of MitraClip placements between 2016 and 2021, a general rising trend was observed (Figure 1).

The yearly total of patients undergoing MitraClip placement (2016–2021) is shown in Figure 1.

In-hospital mortality has been generally low (1–3%) and the rates have not changed over the years statistically (Figure 2).

There has been a general increase in the mean of total hospital charges over the years, with a corresponding declining trend in mean length of hospital stay (Figure 3).

We identified 4918 Index Admissions for MitraClip placement between the months of January and September of 2021. In this Index Admission population, in the 90 days following discharge for Mitraclip placement, there were 780 readmission events. Among these readmission events, 206 events were for heart failure-related readmissions. There were 10 recorded readmissions for clip failure. Major readmission causes in the 90 days following MitraClip placement are summarized in Figure 4.

We analyzed patient comorbidities that held significant association for heart failure readmissions in the 90 days following discharge for MitraClip procedures. Results are summarized in Table 2. Forest plots for association for the MitraClip are summarized in Figure 5. Atrial fibrillation, pulmonary hypertension, and chronic lung disease were conditions that held significant association with 90-day heart failure readmissions after MitraClip.

## 4. Discussion

Secondary Mitral Regurgitation (SMR) differs from primary mitral regurgitation in pathophysiology as well management. Analyzing current literature, SMR could be considered more as a heart failure complication and less of a valvular pathology. Whether the valvular regurgitation or the volume overload related to heart failure progression is the primary driver of the symptoms is quite difficult to distinguish. The two major clinical trials that studied MitraClip interventions along with GDMT, namely the MITRA-FR and the COAPT trials, had conflicting results. Further substudies proved that although the studies compared similar outcomes, the population included in the MITRA-FR trial differed considerably from the patient pool of the COAPT trial. The MITRA-FR trial recruited patients with lower values for the Effective Regurgitant Orifice Area (EROA), and higher values for the Left Ventricular End Diastolic Volume (LVEDV), while the opposite was seen in patients recruited in the COAPT trial. LVEDV is more a consequence of volume overload related to contractile dysfunction, while the EROA is a marker of regurgitation severity. This leads to a concept for disproportionate mitral regurgitation [10]. While the clip placement can be an effective treatment strategy for reducing the severity of MR, the ongoing cardiac remodeling and increase in end diastolic pressures can render the patient symptomatic. However, the selection of optimal patients for the MitraClip still is a topic of debate with general paucity in literature. Although a ratio between EROA and LVEDV could be utilized as a predictor of outcomes with a higher ratio predictive of symptomatic improvement, a clear distinction between these parameters would not always hold true. A higher LVEDV could be a consequence of LV volume overload and eccentric dilation, and a higher EROA could be a consequence of annular dilatation due to LV remodeling. A secondary analysis of the COAPT trial by Lindelfeld et al. [11] analyzed the disproportionate MR theory and could not find an association. Another potential difficulty in the preprocedural echocardiographic evaluation of patients is the eccentric nature of the regurgitant jet and potential underestimation of the EROA using the Proximal Isovelocity Surface Area (PISA). Studies analyze the utility of the Vena Contracta area using 3D models in the quantification of the regurgitant jet [12].

With all the above considerations, the success rate of the MitraClip in symptomatic improvement of heart failure symptoms is remarkable and is widely utilized since its FDA approval in 2013. From 2013 to 2016, there was an 84.4% annual increase in the number of clip placements [13]. The upgoing trend was sustained, with 2488 clip placements in 2021 compared to 869 in 2016 (Table 1, Figure 1). The in-hospital mortality rates related to the procedure have been generally noted to be low. A retrospective analysis conducted by a research data center in Germany noted a 3.4% mortality (all-cause) related to 13,575 MitraClip placements between 2011 and 2015 [14]. Similar trends were observed with the in-hospital mortality between 2016 and 2021, where all-cause mortality remained between 1 and 3%, with a stable trend over the years (Table 1, Figure 2). The mean total hospital charges had a general increasing trend between 2016 and 2021, with a steady decrease in trends observed with the mean length of hospital stay. The mean age for patients undergoing the procedure remained between 77 and 79, with no difference in trends across the years. Sex stratification analysis proved a 50–50 trend in clip placements. Medicare was the primary copayer in 85% of the patients that underwent MitraClip placements between 2016 and 2021.

Analyzing the readmission events in the 90 days following discharge, heart failure was the most common readmission cause, identified in 206 patients out of the 4918 index admission events for MitraClip placements in 2021. A total of 780 readmission events (15.86%) were noted, with heart failure being the cause of readmission in 26.41% of the readmission events. After multivariate regression analysis, atrial fibrillation, pulmonary hypertension, and chronic disease were found to be the comorbidities that held significant association with heart failure readmissions after MitraClip placement. In a retrospective analysis by Kebler et al. [15], baseline troponin levels and preprocedural NYHA functional class were found to be independent predictive factors for heart failure-related readmissions. The study also proved a 65% decrease in the annual rate of hospitalization after MitraClip placements. In another retrospective study by Polimeni et al. [16], preprocedural Left Ventricular End Diastolic Volume Index (LVEDVi) was found to be an independent predictor of worse outcomes, in terms of heart failure hospitalizations and mortality in the one year following clip placement. We identified 10 readmission events where repeat MitraClip placement was due to clip failure. Obesity and uncontrolled hypertension differed between the symptom-worsening group and -improvement group in the ANOVA but was not found to have a significant association with heart failure readmissions in the multivariate regression analysis.

While annular dilatation due to LV dysfunction, causing subsequent tethering of the posterior MitraClip, is the most common mechanism in secondary MR, the annual dilatation caused due to LA dilatation is also a known pathophysiologic mechanism [17]. This can potentially explain the higher association for worsening outcomes in secondary MR patients, even after clip placement. Documented diagnosis of atrial fibrillation held a significant association with heart failure-related readmissions, with an odds ratio of 3.77 (1.82–4.23). In a meta-analysis by Shah et al. [18], atrial fibrillation was associated with higher procedural complications as well as readmissions for heart failure and bleeding-related complications after the clip placement. In a subanalysis of the COAPT trial by Gertz et al. [19], patients with a history of AF had larger left atrial and mitral valve orifice areas with higher LV ejection fraction and smaller LV volumes, suggesting an atrial mechanism contribution to functional MR. Chronic Lung Diseases were found to have a 1.91 (1.32–2.77) higher association with heart failure-related admissions following the MitraClip. The impact of COPD on MitraClip outcomes have been analyzed in the past. In a substudy of the COAPT patient pool by Saxon et al. [20], documented diagnosis of COPD attenuated the survival benefits after MitraClip placements when compared to GDMT alone; however, no association was found between one-year heart failure hospitalization. Documented diagnosis of pulmonary hypertension had a 3.96 (1.49–5.55) higher odds of association with heart failure-related readmissions (Table 2, Figure 5). In a study involving 643 patients in the TRAnscatheter Mitral valve Interventions (TRAMI) registry [21], it was noted that patients with severe elevations in the pulmonary artery systolic pressures had higher mortality after MitraClip placement and that the safety and efficacy of MitraClip therapy was proved in patients with even advanced stages of pulmonary hypertension.

As mentioned in the introduction section, the optimal patient selection for MitraClip (TEER) is a gray area. Based on the results of the MITRA-FR and the COAT trial, there have been ongoing studies based on the disproportionate MR theory and analyzing the predictive value of EROA/LVEDV ratio in patient outcomes. Use of the Vena Contracta area as an eccentric nature of the jet underestimating EROA through PISA is also considered to optimize patient selection for the same. Considering the factors that had significant association with heart failure readmissions, chronic lung diseases were found to have significant association. This included COPD, asthma, and interstitial lung pathologies together. The likely explanation would be the worsening diastolic function and increase in pulmonary artery pressures further worsening the remodeling, leading to further regurgitation. As per the current guidelines from AHA/ACC, the pulmonary artery systolic pressure has been used as a cut-off above which TEER is not tried. However, based on our analysis, pulmonary hypertension of any severity was found to be significantly associated with heart failure readmissions. Atrial fibrillation is yet another factor that was found to have significant association with heart failure readmissions. Longstanding atrial fibrillation can be the cause of functional MR through Carpentier I mechanism of mitral annular dilatation due to left atrial enlargement. The predictive factors for heart failure readmissions need to be further studied in prospective analysis, as advanced heart failure therapies like LVADs and heart transplants are getting more robust, and patients who are unlikely to benefit from TEER could be considered for other options like continuous ionotropic support and mechanical circulatory support of even heart transplants.

## 5. Conclusions

Between 2016 and 2021, there was a 186.36% increase in the total number of clip placements, with 2488 total procedures in 2021 compared to 869 in 2016. There has been a general increasing trend in the mean of total hospital charges and decreasing trend in the mean length of hospital stay, 4.94 (4.47–5.41) days in 2016 compared to 4.03 (3.71–4.34) days in 2021. Atrial fibrillation, pulmonary hypertension, and diagnosis of chronic lung diseases were factors associated with heart failure readmissions in the 90 days following the MitraClip procedure.

## 6. Limitations of the Study

Although National Inpatient sample and National Readmission databases are the largest publicly available databases in the United States, the databases are not without limitations. The echocardiographic parameters are not included in the admissions that underwent MitraClip placement, nor could the patients be followed up longitudinally to assess medication adherence. NRD gives an advantage of longitudinally following patients, but factors like medication adherence (GDMT) are hard to assess and account for in confounding due to the nature of the databases. The diagnoses used in the patient selection are based on ICD 10 or PCS codes, which are also subject to inter-provider variability.

## Figures and Tables

**Figure 1 medsci-13-00081-f001:**
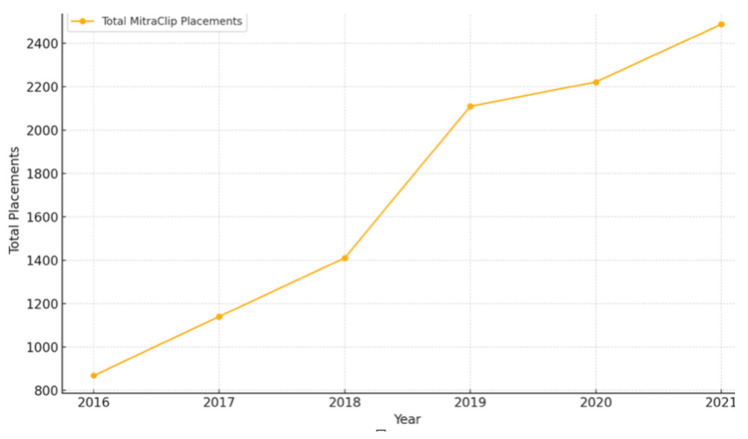
Trends in MitraClip Placements (2016–2021).

**Figure 2 medsci-13-00081-f002:**
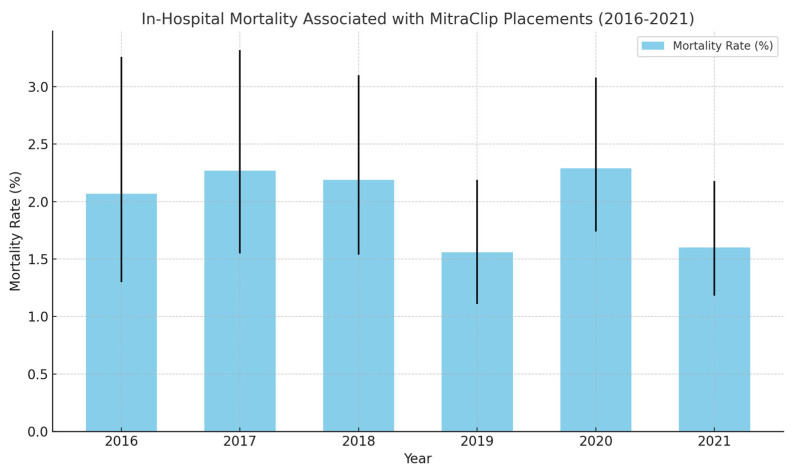
Trends in in-Hospital Mortality (2016–2021).

**Figure 3 medsci-13-00081-f003:**
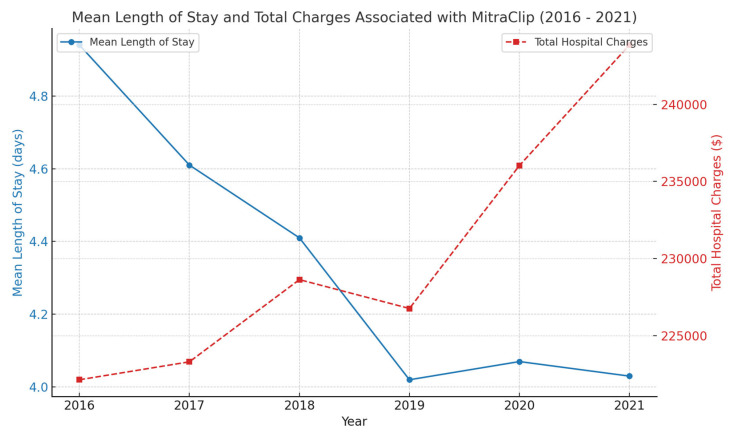
Trends in Mean Length of Hospital Stay and Mean of Total Hospital Charges (2016–2021).

**Figure 4 medsci-13-00081-f004:**
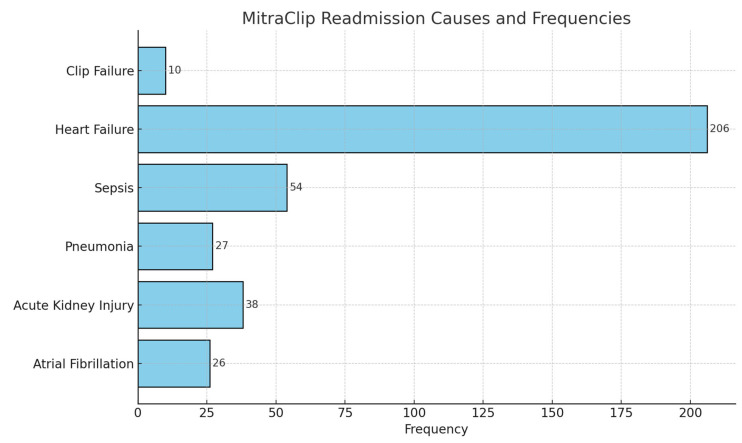
Major Readmission Events in the 90 Days Following MitraClip Placement.

**Figure 5 medsci-13-00081-f005:**
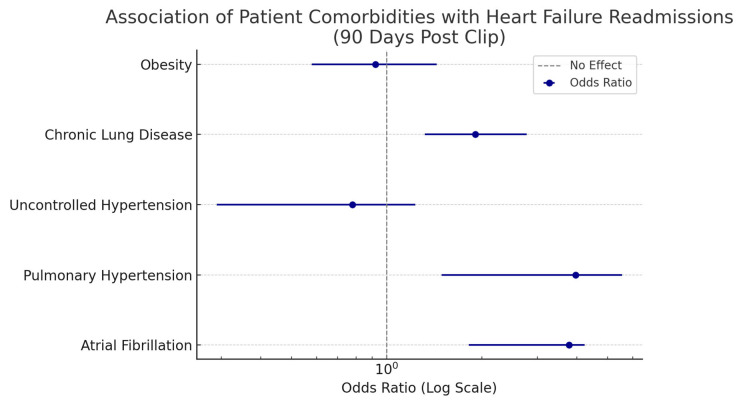
Association of Comorbid Conditions with Heart Failure Readmissions.

**Table 1 medsci-13-00081-t001:** Socioeconomic and Healthcare resource utilization trends for MitraClip procedures (2016–2021).

		**2016**	**2017**	**2018**	**2019**	**2020**	**2021**
Total Number of MitraClip placements		869	1141	1411	2110	2222	2488
Mean Age		77.71 (77.05–78.41)	78.58 (78.02–79.14)	77.79 (77.72–78.31)	76.95 (76.51–77.39)	76.13 (75.69–76.58)	76.19 (75.78–76.59)
Sex (%)	Male:	50.06	52.05	54.07	54.69	55.65	54.09
	Female:	49.94	47.94	45.92	45.38	44.34	45.90
Mean Hospital Length of Stay (days)		4.94 (4.47–5.41)	4.61 (3.89–5.31)	4.41 (4.05–4.84)	4.02 (3.73–4.32)	4.07 (3.77–4.37)	4.03 (3.71–4.34)
Mean of Total Hospital Charges ($)		222,130.8	223,295.2	228,626.9	226,755.4	236,031.1	243,849.8
In-Hospital Mortality (%)		2.07 (1.30–3.26)	2.27 (1.55–3.32)	2.19 (1.54–3.10)	1.56 (1.11–2.19)	2.29 (1.74–3.08)	1.60 (1.18–2.18)
Race (%)	White	79.47	81.80	80.46	79.25	78.25	78.96
	Black	9.13	7.26	7.26	9.74	10.04	9.88
	Hispanic	6.06	6.62	7.26	5.60	5.97	5.89
Primary Insurance Payer (%)	Medicare	84.60	88.74	87.45	84.60	82.65	84.60
Median Household Income (%)	1st quartile:	25.55	21.55	20.79	22.77	22.76	22.46
	2nd quartile:	24.04	23.33	24.44	23.11	27.24	22.59
	3rd quartile:	26.25	28.32	27.46	28.65	25.41	27.50
	4th quartile:	24.15	26.80	27.31	25.47	24.59	24.25

**Table 2 medsci-13-00081-t002:** Association of Comorbid Conditions with 90-Day Heart Failure Readmission Events.

Patient Comorbidities	Association with Heart Failure Readmissions: MV Clip Group (Odds Ratio with 95% Confidence Interval)
Atrial Fibrillation	3.77 (1.82–4.23)
Pulmonary Hypertension	3.96 (1.49–5.55)
Uncontrolled Hypertension	0.78 (0.29–1.23)
Chronic Lung Disease	1.91 (1.32–2.77)
Obesity	0.92 (0.58–1.44)

## Data Availability

Data publicly available in the HCUP NIS/NRD database.

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
