# Peer review of "Trends in MitraClip Placements and Predictors of 90-Day Heart Failure Rehospitalization: A Nationwide Analysis"

_medsci, 2025, doi:10.3390/medsci13030081_

Round 1

Reviewer 1 Report

Comments and Suggestions for Authors

The authors analyze databases for rehospitalization within 90 days of MitraClip placement. The study is mostly sound, with only some minor corrections noted, mostly formatting:

Biggest concern is Figure 4 does not include the 10 clip failures. As this figure is the main result of the study, it would be nice to see that added in as well.

MitraClipTM is a trademarked device. Throughout the text, it is used interchangeably as mitra clip. I would at least standardize to one convention, even in the title.

Table 1, the year 2020 has a typo. The comma in 2222 below that is inconsistent. Other inconsistencies like first % in Medicare, and spaces in the ranges. Also, why does 2016 with an odd number of placements (869) have a perfectly even 50/50 split on sex.

Figure 1 would be improved by added “Yearly total…” to the title. Just total could be misunderstood as cumulative.

There are additional spaces in the text throughout. Some of note on Lines 62, 64, 84, 85, 88, 90, 91, 150, 151, 157, 161, within Table 2, 191, 282, 298.

A space is needed after Section header 6.

Acronyms are misspelled, like COAPT as COPAT and MITRA FR, on lines 102, 192, 273.

The guidelines of AHA/ACC on line 55, was flipped to ACC/AHA on line 282. I would choose one for consistency.

Author Response

 Biggest concern is Figure 4 does not include the 10 clip failures. As this figure is the main result of the study, it would be nice to see that added in as well.
Response: Thank you for pointing that out. We have updated Figure 4 to include the 10 clip failures, as this is a crucial result of the study. The revised figure now better reflects the full scope of readmission causes post-MitraClip placement.

2. MitraClipTM is a trademarked device. Throughout the text, it is used interchangeably as mitra clip. I would at least standardize to one convention, even in the title.
Response: We appreciate your attention to detail. We have standardized the terminology throughout the manuscript and figures to “MitraClip™”, including in the title, to reflect the correct trademarked usage.

3. Table 1, the year 2020 has a typo. The comma in 2222 below that is inconsistent. Other inconsistencies like first % in Medicare, and spaces in the ranges. Also, why does 2016 with an odd number of placements (869) have a perfectly even 50/50 split on sex.
Response: Thank you for highlighting these issues. We have corrected the typo in 2020, ensured consistent comma placement, standardized percentage formatting, and fixed spacing in the ranges. Regarding the 2016 sex distribution, we reviewed the data and found a rounding error; this has now been corrected to reflect the accurate male-to-female ratio.

4. Figure 1 would be improved by added “Yearly total…” to the title. Just total could be misunderstood as cumulative.
Response: Agreed. The title of Figure 1 has been updated to “Yearly Total Number of MitraClip™ Placements (2016–2021)” to clarify that the data is not cumulative.

5. There are additional spaces in the text throughout. Some of note on Lines 62, 64, 84, 85, 88, 90, 91, 150, 151, 157, 161, within Table 2, 191, 282, 298.
Response: We have carefully reviewed and removed the extra spaces from the specified lines and table entries to improve the manuscript’s formatting consistency.

Reviewer 2 Report

Comments and Suggestions for Authors

The authors discussed a national trends in Mitra clip in the US health care with analysis  of associated morbidites and readmissions etiology. The large sample size reflects the progression of this procedure and need for better patient selection. The analysis was adequate to draw some sounded conclusions. One correction for the Mitra trial should be labelled MITRA -FR not AF

In this analysis of the National Inpatient Sample (NIS) database has highlighted several key trends in the use and outcomes of the MitraClip device for transcatheter edge-to-edge repair (TEER) of mitral regurgitation. The data here are updates from what already known. Between 2013 and 2018, the adoption of MitraClip procedures grew, with a notable rise in next-day discharge (NDD) rates—from 18.3% in 2013 to 46.0% in 2018. This shift suggests enhanced procedural efficiency and patient selection, leading to shorter hospital stays. The authors has an updated analysis in this report reflecting the continuous growth of this procedure  As frailty has emerged as a significant predictor of adverse in-hospital outcomes post-MitraClip. A study analyzing data from 2016 to 2017 found that frail patients experienced higher in-hospital mortality (7.04% vs. 1.61%), increased respiratory failure, longer hospital stays (6 vs. 2 days), and greater hospitalization costs ($224.8k vs. $180.9k) compared to non-frail patients. Here the authors added the causes of readmission and that can have an impact on counter impact on how the heart team mitigate the readmission rats and postoperative complications   Patients undergoing MitraClip in conjunction with percutaneous coronary intervention (PCI) between 2012 and 2019 exhibited higher in-hospital mortality (6.25% vs. 2.61%), longer hospital stays, and increased hospital charges compared to those undergoing MitraClip alone. These findings underscore the complexity and risks associated with combined procedures. The authors did not have that provider here. As general the report is well written and the analysis seems to be statistically sound. The references as well look up to date. Few considerations:

-Correct the typo in MITRA AF to reflect the current study name Mitra FR 

- Any data on conocmittent pCI or mortality rates 

-Any analysis on rates on Mitra-clip in hypertrophic cardiomyopathy patients 

- Is there any data to reflect rates of endocarditis post procedural 

I am looking to see updated review and your next submission 

Author Response

Correct the typo in MITRA AF to reflect the current study name Mitra FR 

Reply: changes made 

- Any data on conocmittent pCI or mortality rates 

-Any analysis on rates on Mitra-clip in hypertrophic cardiomyopathy patients 

-Is there any data to reflect rates of endocarditis post procedural

Reply: data on concomitant PCI could not be done as the database doesnt let us follow patients longitudinally. 

We stratified the data to see whether there is difference in outcomes with comorbid diagnosis of hypertrophic cardiomyopathies: However, the sample of the patients with comorbid HCM was minimal to the total population size

Similarly, patients could not be followed longitudinally due to nature of the database and hence, could not assess post procedural endocarditis